# UUKG: Unified Urban Knowledge Graph Dataset for Urban Spatiotemporal Prediction

**Yansong Ning[1], Hao Liu[1,2]\*, Hao Wang[3], Zhenyu Zeng[3], Hui Xiong[1,2]**

[1] AI Thrust, The Hong Kong University of Science and Technology (Guangzhou)
[2] CSE, The Hong Kong University of Science and Technology
[3] Alibaba Cloud Intelligence Group
`yning092@connect.hkust-gz.edu.cn` `liuh@ust.hk`
`cashenry@126.com` `zhenyu.zzy@alibaba-inc.com` `xionghui@ust.hk`

## Abstract

Accurate Urban SpatioTemporal Prediction (USTP) is of great importance to the development and operation of the smart city. As an emerging building block, multi-sourced urban data are usually integrated as urban knowledge graphs (UrbanKGs) to provide critical knowledge for urban spatiotemporal prediction models. However, existing UrbanKGs are often tailored for specific downstream prediction tasks and are not publicly available, which limits the potential advancement. This paper presents UUKG, the unified urban knowledge graph dataset for knowledge-enhanced urban spatiotemporal predictions. Specifically, we first construct UrbanKGs consisting of millions of triplets for two metropolises by connecting heterogeneous urban entities such as administrative boroughs, POIs, and road segments. Moreover, we conduct qualitative and quantitative analysis on constructed UrbanKGs and uncover diverse high-order structural patterns, such as hierarchies and cycles, that can be leveraged to benefit downstream USTP tasks. To validate and facilitate the use of UrbanKGs, we implement and evaluate 15 KG embedding methods on the KG completion task and integrate the learned KG embeddings into 9 spatiotemporal models for five different USTP tasks. The extensive experimental results not only provide benchmarks of knowledge-enhanced USTP models under different task settings but also highlight the potential of state-of-the-art high-order structure-aware UrbanKG embedding methods. We hope the proposed UUKG fosters research on urban knowledge graphs and broad smart city applications. The dataset and source code are available at https://github.com/usail-hkust/UUKG/.

## 1 Introduction

Urban SpatioTemporal Prediction (USTP) aims to forecast future urban dynamics by simultaneously capturing the spatial and temporal autocorrelations from past observations. Recently, with the development of machine learning technologies and the collection of large-scale urban data, USTP has achieved remarkable progress in various urban predictive tasks, such as traffic management, pollution monitoring, and emergency response [1, 2, 3]. USTP has become an essential service of the modern smart city.

In prior literature, many efforts have been devoted to improving the USTP performance by exploiting latent knowledge from diverse urban data sources [4]. In particular, one commonly used approach is manually extracting features from different datasets to integrate additional information.

---

\*Corresponding author.

37th Conference on Neural Information Processing Systems (NeurIPS 2023) Track on Datasets and Benchmarks.

For example, Tompson *et al.* [5] and Chen *et al.* [6] customize urban features from external dataset for crime prediction and bike trip prediction. However, these approaches heavily rely on a deep understanding of the application domain and are labor-intensive. Recently, inspired by the success of the Knowledge Graph (KG) in natural language processing tasks, the urban knowledge graph has been adopted to improve USTP. For instance, Wang *et al.* [7]

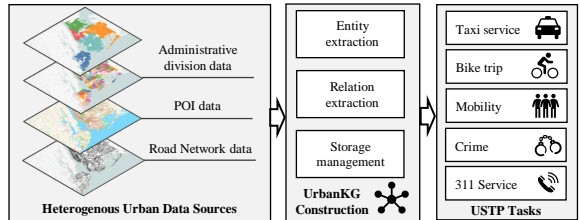

Figure 1: The generation process of UUKG.

construct a dedicated spatiotemporal knowledge graph by regarding trajectory and timestamp as entities to improve trajectory prediction. Liu *et al.* [8] construct user check-in relations to help mobility prediction. However, existing UrbanKGs are **task-specific and none of them is publicly available**, which discourages researchers from adopting it for their own work and thus limits the flourishing of knowledge-enhanced urban spatiotemporal prediction.

Therefore, an open-sourced and multifaceted urban knowledge graph dataset compatible with various USTP tasks is necessary to be proposed. It is a non-trivial problem due to the following two challenges. (1) *How to construct unified urban knowledge graphs?* The urban data collected from different devices and providers describe the urban system from different aspects and granularities. They are usually disjoint datasets with different spatiotemporal ranges, which cannot be directly joint utilized for USTP. It is appealing to extract and align heterogeneous urban knowledge in a **unified graph organization** to satisfy diverse downstream USTP requirements. (2) *How to preserve complicated structural urban knowledge?* Existing UrbanKG-driven USTP approaches simply adopt general KG embedding methods or design complex task-specific module to project symbolic entities and relations into low-dimensional embeddings, which **fails to preserve unique structural patterns** in the urban knowledge graph. As illustrated in Figure 2, the urban knowledge graph includes diverse structures such as hierarchy and cycle. It is crucial to capture such high-order semantic knowledge to empower downstream spatiotemporal prediction tasks.

To address the aforementioned challenges, in this study, we present an Unified Urban Knowledge Graph (UUKG) dataset for knowledge-enhanced urban spatiotemporal predictions. Figure 1 illustrates the workflow of UUKG construction. For a given city, we first construct an Urban Knowledge Graph (UrbanKG) from multi-sourced urban data. By extracting and organizing entities (*e.g.*, POIs, road segments, *etc.*) into a multi-relational heterogeneous graph, UrbanKG encodes various high-order structural patterns in a unified configuration (*i.e.*, a multi-scale spatial hierarchy), which facilitates joint processing for various downstream USTP tasks. Moreover, we **qualitatively and quantitatively analyze these diverse high-order structures** (*i.e.*, hierarchies and cycles in Figure 2) to guide us in using tailored KG embedding methods to derive effective and generalizable knowledge representations. By learning **structure-aware embeddings** of entities and relations using state-of-art non-Euclidean space embedding models (*e.g.*, modeling hierarchies in hyperbolic space and capturing cycles in spherical space), the high-order structure information could be preserved in a proper way.

Finally, we conduct comprehensive experiments to benchmark diverse urban tasks, including the UrbanKG completion task, three urban flow prediction tasks, and two urban event prediction tasks. The empirical studies not only validate the effectiveness of the unified urban knowledge graph for improving various USTP tasks, but also uncover the high-order structures within UrbanKG, with proper modeling, can further strengthen the urban spatiotemporal prediction performance.

Our contributions are summarized as follows:

- We propose UUKG, **the first open-source UrbanKG dataset** for knowledge-enhanced urban spatiotemporal predictions. As a pilot study, we envision our experiences and results offering exciting opportunities to advance previous USTP methods.

- We demonstrate **the importance of urban high-order structure modeling** for urban knowledge representation and investigate to use structure-aware KG embedding methods to capture them effectively.

- Extensive experiments on KG completion and five USTP tasks demonstrate the effectiveness of the constructed UrbanKG and verify modeling high-order structures is beneficial to urban spatiotemporal prediction.

Table 1: The statistics of raw datasets.

| Dataset | Description | Sample format | Records | |
|---|---|---|---|---|
| | | | New York | Chicago |
| Administrative division | Boundary | *["New Yorck", range(40.50, 40.91, -74.25, -73.70)]* | 1 | 1 |
| | # of Borough | *["Queens", polygon(40.54 -73.96, ...)]* | 5 | 6 |
| | # of Area | *["Jamaica", "Queens", polygon(40.69 -73.82, ...)]* | 260 | 77 |
| Road network | # of Segment | *[road id, road name, start junction, end junction, type, line range]* | 110,919 | 71,578 |
| | # of Category | *[191751, "Queens Boulevard", 59378, 4798, tertiary, line(40.73 -73.82, ...)]* | 5 | 5 |
| | # of Junction | *[junction id, junction type, coordinate]* | 62,627 | 37,342 |
| | # of Category | *[59378, crossing, coordinate(40.78 -73.98)]* | 5 | 6 |
| POI | # of POI | *[poi id, poi name, poi type, coordinate]* | 62,450 | 31,573 |
| | # of Category | *[34633854, "Empire State Building", corporation, coordinate(40.75, -73.99)]* | 15 | 15 |

## 2 UrbanKG Construction

We first introduce the datasets that will be used for urban knowledge graph construction and data preprocessing details, then we present the construction of urban knowledge graphs.

### 2.1 Data Collection and Preprocessing

We acquire urban knowledge for two large cities, New York and Chicago, from three data sources. Table 1 summarizes the statistics of the datasets.

**Administrative Division data**. The administrative division describes multi-level spatial entities that the government uses to make administrative decisions, which contains rich geographical information for USTP task [9, 10]. In this work, we collect three administrative divisions, *i.e.*, city, borough, and area. We first manually define the rectangular latitude and longitude boundaries for each city (*e.g.*, New York and Chicago). Then we collect borough and area data from NYC Gov [2] and CHI Gov [3], the official websites of New York and Chicago. Each borough record contains a borough name and polygon boundaries. Each area record contains an area name, corresponding borough name, and the polygon boundaries. Example in Table 1 is a record of *"Jamaica"* area in the *"Queens"* borough.)

**Road Network Data**. The road network provides rich topology knowledge of the transportation system, which plays a critical role in various USTP tasks (*e.g.*, traffic congestion prediction [11] and traffic incident detection [12]). We utilize the rectangular of each city to query the corresponding road network data (road segments and road junctions) from Open Street Map (OSM [4]). Each road segment record contains a unique id, a road name, a start road junction id, a end road junction id, and line geographical range. Each road junction record contains a unique id, junction type and coordinate (latitude and longitude). Example in Table 1 is a road record of *"Queens Boulevard"*, and it starts from junction *594778* and ends at junction *4798*.

**POI Data**. POI data contains urban contextual semantics, which have been widely adopted as auxiliary features to enhance USTP tasks (*e.g.*, traffic prediction[13] and crime prediction [14]). We collect POI data from OSM. We utilize the boundary of each city from the administrative division data to query the corresponding POI records through the open API provided by OSM. Each POI record consists of a unique id, a POI name, POI category and the coordinate. Example in Table 1 is a POI record where *(40.75, -73.99)* is the location and the POI is a *corporation*.

Before constructing the UrbanKG, we first preprocess the raw datasets. We filter out POIs and road networks that don't belong to any administrative borough or area in each city. Besides, we merge minority POI categories (*e.g.*, grandstand and canopy) to avoid potential long-tail issues. As records from different data sources are disjoint, we align each record according to the administrative areas.

### 2.2 Knowledge Graph Construction

Then we introduce the process of constructing the urban knowledge graph (UrbanKG).

**Definition 1** *UrbanKG. The UrbanKG is defined as a multi-relational graph $\mathcal{G} = (\mathcal{E}, \mathcal{R}, \mathcal{F})$, where $\mathcal{E}$, $\mathcal{R}$ and $\mathcal{F}$ is the set of urban entities, relations and facts, respectively. In particular, facts are*

---

[2]https://www.nyc.gov/

[3]https://www.chicago.gov/

[4]https://www.openstreetmap.org/

Table 2: Major relations in UrbanKG.

| Relation | Symmetric | Head & Tail Entity | Abbrev. | Relation | Symmetric | Head & Tail Entity | Abbrev. |
|---|---|---|---|---|---|---|---|
| POI Locates at Area | × | (POI, Area) | PLA | Junction Belongs to Road | × | (Junction, Borough) | JBR |
| Road Locates at Area | × | (Road, Area) | RLA | Borough Nearby Borough | ✓ | (Borough, Borough) | BNB |
| Junction Locates at Area | × | (Junction, Area) | JLA | Area Nearby Area | ✓ | (Area, Area) | ANA |
| POI Belongs to Borough | × | (POI, Borough) | PBB | POI Has POI Category | × | (POI, PC) | PHPC |
| Road Belongs to Borough | × | (Road, Borough) | RBB | Road Has Road Category | × | (Road, RC) | RHRC |
| Junction Belongs to Borough | × | (Junction, Borough) | JBB | Junction Has Junction Category | × | (Junction, JC) | JHJC |
| Area Locates at Borough | × | (Area, Borough) | ALB | - | - | - | - |

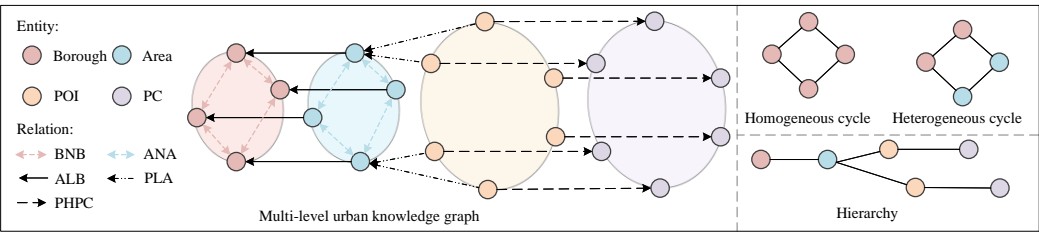

Figure 2: An illustrative example of the urban knowledge graph with diverse hierarchical and cyclic structure. There are four types of entities (Borough, Area, POI and POI Category (PC)) and five types of relations (BNB, ALB, ANA, PLA and PHPC).

*defined as $\mathcal{F} = \{\langle h, r, t \rangle \mid h, t \in \mathcal{E}, r \in \mathcal{R}\}$, where each triplet $\langle h, r, t \rangle$ describes that head entity $h$ is connected with tail entity $t$ via relation $r$. For example, $\langle Queen, Nearby, Brooklyn \rangle$ represents the fact that Queen and Brooklyn are geographically adjacent.*

The UrbanKG encodes diverse urban semantic knowledge such as the multi-level spatial adjacency (*e.g.*, the Hammels area in Queens borough) and ontology (*e.g.*, category of POIs and roads). We detail the entity and relation extraction below.

### 2.2.1 Entity Extraction

We extract 8 types of entities for UrbanKG: (1) **Administrative Borough (Borough).** Borough describes high-level administrative boundaries of a city. For example, New York has five boroughs: Queen, Bronx, Brooklyn, Manhattan, and Staten Island. (2) **Functional Area (Area).** Area refers to the subdivision of a city according to its function, such as residential, industrial, and commercial areas. (3) **Point-Of-Interest (POI).** A POI is a basic functional unit and venue. For example, cinemas and hospitals are two common types of POIs and they are the places people often check in. (4) **Road Segment (Road).** Road is the key element in the city and it forms the traffic network, which provides a reference and basis for human mobility. (5) **Road Junction (Junction).** Junction is where two or more road segments intersect. It is important for the urban road network as they have a great impact on traffic capacity and safety. (6) **POI Category (PC).** POI category describes the property and function of POI, such as finance, catering and so on. We define 15 categories of POI including finance, parking area, shopping, catering, and so on. (7) **Road Category (RC).** Road category describes the features of the road segment, such as motorway, trunk and so on. We preserve the six-types of most frequent road segments including motorway (expressway or river-crossing tunnels), primary traffic road, secondary traffic road, tertiary traffic road, residential traffic road, and trunk (branch roads such as expressway outbound bypass roads). (8) **Junction Category (JC).** Junction category helps distinguish several special types of road junctions, such as roundabout. We keep five types of road junction including motorway junction (road junction in expressway or river-crossing tunnel), traffic signal (road junction having traffic light), turning circle (road junction which is a roundabout), stop (road junction having stop signal) and crossing (road junction with no special type).

We report the detailed statistics of entities in Appendix A.1.

### 2.2.2 Relation Construction

As shown in Table 2, we define 13 relations for UrbanKG:

Table 3: Statistics of two UrbanKGs. We report graph hyperbolicity values $\delta$ (always greater than or equal to zero while lower means more hierarchical) and the number of cycles.

| Dataset | # Entity | # Relation | # Triplet | # Train | # Valid | # Test | # Cycle | # Hyperbolicty |
|---|---|---|---|---|---|---|---|---|
| NYC | 236,287 | 13 | 930,240 | 837,216 | 46,512 | 46,512 | 1,090,884 | $\delta = 0$ |
| CHI | 140,602 | 13 | 564,400 | 507,960 | 28,220 | 28,220 | 532,108 | $\delta = 0$ |

**Geographic Inclusion.** This type of relation describes the geographical inclusion relations between entities. We extract 5 geographical inclusion relations for UrbanKG: *(1) POI/Road/Junction Locates at Area (PLA/RLA/JLA).* Through these relations, we can obtain the fact between Area and POI/Road/Junction. For example, triplet ⟨*POI 32, PLA, Area 75*⟩ indicates that POI 32 is located in Area 75. *(2) POI/Road/Junction Belongs to Borough (PBB/RBB/JBB).* Through these relations, we can obtain the fact between Borough and POI/Road/Junction. *(3) Area Locates at Borough (ALB).* Through this relation, we can obtain the geographic inclusion relations between Area and Borough. For example, triplet

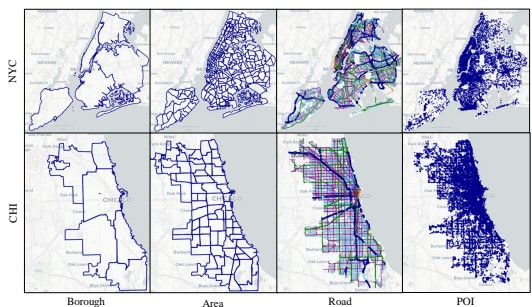

Figure 3: An illustrative visualization of the four UrbanKG entities in NYC and CHI.

⟨*Area 75, ALB, Borough 1*⟩ explains that Area 75 belongs to Borough 1. *(4) Junction Belongs to Road (JBR).* Through this relation, we can describe the Junction on which Road, thus could preserve the topology of the road network.

**Geographic Aadjacency.** This type of relation describes the geographical adjacency relations between entities. We extract 2 relations of this type for UrbanKG: *(1) Borough Nearby Borough (BNB).* Through this relation, we can obtain the adjacency fact between Boroughs. *(2) Area Nearby Area (ANA).* Through this relation, we describe the adjacency between Areas.

**Category.** This type of relation connects entities to the category they belong to. We extract three category relations for UrbanKG: *(1) POI Has POI Category (PHPC). (2) Road Has Road Category (RHRC). (3) Junction Has Junction Category (JHJC).* For example, triplet ⟨*POI 32, PHPC, PC 4*⟩ indicates POI 32 belongs to the category PC 4.

### 2.2.3 UrbanKG Management

Following the above process, we obtain UrbanKGs for two large cities as shown in Table 3, *i.e.*, New York (NYC) and Chicago (CHI). To serve the large-scale knowledge graph with millions of triplets, we adopt Neo4j, a graph database system, to store, query, and update the UrbanKG. The data privacy issue is critical in smart city applications. In this study, the dataset we use doesn't include any user-level information. However, recent studies prove that sensitive individual information can be inferred from a few location check-ins [15]. Thus, we round the coordinate of each record to 100 meters to avoid potential information leakage when incorporating UrbanKG with other urban spatio-temporal data.

### 2.3 Visualization and Structural Analysis

As shown in Figure 3, the multi-level UrbanKG contains entities providing administrative division information (Borough & Area), traffic topology (Road network), and urban function distribution (POI), and they together form rich semantic information (*i.e.*, High-order hierarchy and high-order cycle) about the city.

**High-order Hierarchy.** Hierarchical information is ubiquitous in capturing knowledge from KG [16, 17, 18] since much knowledge is commonly organized hierarchically. For example, the hierarchy (*Borough->Area->POI->PC*) in Figure 2 organizes urban knowledge in a hierarchical way. Embracing and modeling such hierarchy empowers to uncover deeper semantics, *e.g.*, we can infer the main function of an area from the POI quantities and categories in this area. As shown in Table 3, we also quantitatively find the hyperbolicities [19] of two UrbanKGs are zero, which means that UrbanKGs

are not significantly different from a tree in terms of their geometric properties [20, 21]. Thus they are suitable to be represented in hyperbolic space [22, 19], where hierarchies can be losslessly embedded.

**High-order Cycle.** Cycle modeling is beneficial to improve knowledge representation [23, 24, 25] as it implies rich semantics of graphs. We find their existence in constructed UrbanKG by analyzing the topological structure. For example, the homogeneous cycle and heterogeneous cycle in Figure 2 uncover *Borough* geographic semantics and the geographic semantics of *Area* and *Borough*, respectively. Moreover, Table 3 counts the number of cycles in the two UrbanKG to further verify our analysis. Unlike hierarchy, these cycles are suitable to be modeled in spherical space [22, 26].

The qualitative and quantitative analysis reveal diverse high-order structural patterns in UrbanKG, and thus inspires us to leverage state-of-the-art high-order structure-aware knowledge graph embedding methods to extract tailored and comprehensive urban knowledge. For instance, UrbanKG can be embedded into hyperbolic space to capture hierarchies or into spherical space to model cycles.

# 3 Structure-aware UrbanKG Embedding

This section investigates the impact of different structure-aware embedding methods (*e.g.*, the hyperbolic space embedding module to capture hierarchy) on UrbanKG representation learning. We first give problem formulation of UrbanKG embedding and introduce how to utilize structure-aware embedding methods to represent those high-order structures within UrbanKG.

## 3.1 Problem Formulation

Given the constructed UrbanKG $\mathcal{G} = (\mathcal{E}, \mathcal{R}, \mathcal{F})$, we aim to learn low-dimensional entity embeddings $e_{\mathcal{E}}$ and relation embeddings $e_{\mathcal{R}}$, so that diverse high-order structure knowledge can be preserved to benefit various downstream USTP tasks.

## 3.2 Experimental Setup

We compare the performance gap of different structure-aware knowledge graph embedding methods to investigate which approaches are suitable for modeling the unique structures in UrbanKG. The quality of UrbanKG embedding is evaluated using the link prediction task [27], which aims to predict the missing head or tail entity for a triplet $\langle h, r, t \rangle$. For each UrbanKG in Table 3, we follow standard data augmentation protocol [28] by adding inverse relations.

**Evaluation Metrics.** We report two popular metrics: (1) Mean reciprocal rank (MRR), the mean of inverse of correct entity ranking [29]. (2) Hits@K (K=1,3,10), the percentage of correct entities in top-K ranked entities [30, 31]. Note we filter out true triplets appearing in the training set during evaluation [30], because predicting a low rank for those triplets should not be penalized.

**Methods.** We utilize classical Euclidean methods as baselines and choose state-of-art structure-aware non-Euclidean approaches as backbones.

- **Euclidean Models**: (1) TransE [30] is a classical translational model; (2) DisMult [32] is a matrix factorization model; (3) ComplEx [33] is an extension of DisMult in complex space; (4) RotatE [34] is a rotation-based method in complex space; (5) MuRE [16] is a translational distance method with a diagonal relational matrix. (6) TuckER [35] is a tensor decomposition method. (7) QuatE [36] is a hypercomplex space embedding method.

- **Non-Euclidean Models**: (1) MuRP/MuRS [16] are hyperbolic or spherical model with diagonal relational matrix; (2) RotH/RefH [17] are hyperbolic embedding method with rotation or refelection. (3) AttH [17] is a hyperbolic embedding method combining rotation and reflection; (4) ConE [18] is a hyperbolic embedding method with transformation between hyperbolic cone; (5) M2GNN[5] [26] is a GNN-based product space embedding model which can model cycles and hierarchies at the same time. (6) GIE [37] is a product space model considering geometry interaction.

**Implement Details.** We implement all models using PyTorch. All experiments are conducted on eight NVIDIA RTX 3090 GPUs. The negative sampling size is fixed to 50. We conduct hyperparameters grid search by early stop on the validation sets and we report details in Appendix A.2.

---

[5]Due to the large scale of our UrbanKG, we didn't perform Graph Neural Updater when reproducing.

Table 4: Overall link prediction results on NYC and CHI dataset. We utilize E, C, S, H, P to denote the Euclidean, Complex, Spherical, Hyperbolic and Product space, respectively. Best results in each space are underlined.

| Type | Space | Model | NYC | | | | CHI | | | |
|---|---|---|---|---|---|---|---|---|---|---|
| | | | MRR | Hits@10 | Hits@3 | Hits@1 | MRR | Hits@10 | Hits@3 | Hits@1 |
| Euclidean models | E | TransE | .507 | .563 | .528 | .470 | .485 | .556 | .501 | .436 |
| | E | DisMult | .401 | .478 | .433 | .355 | .395 | .479 | .469 | .394 |
| | E | MuRE | .516 | .613 | .545 | .468 | .493 | .601 | .536 | .437 |
| | E | TuckER | .513 | .609 | .541 | .466 | .488 | .584 | .525 | .423 |
| | C | RotatE | .274 | .363 | .309 | .220 | .306 | .385 | .336 | .258 |
| | C | ComplEx | .259 | .357 | .305 | .195 | .304 | .385 | .337 | .253 |
| | C | QuatE | .321 | .388 | .347 | .282 | .396 | .490 | .427 | .345 |
| Non-Euclidean models | S | MuRS | .528 | .622 | .552 | .478 | .501 | .619 | .536 | .437 |
| | H | MuRP | .545 | .635 | .570 | .497 | .519 | .634 | .555 | .456 |
| | H | RotH | .526 | .601 | .550 | .472 | .511 | .626 | .548 | .447 |
| | H | RefH | .524 | .610 | .549 | .473 | .509 | .629 | .542 | .451 |
| | H | ATTH | .539 | .603 | .556 | .503 | .514 | .610 | .542 | .463 |
| | H | ConE | .542 | .629 | .563 | .485 | .513 | .617 | .535 | .465 |
| | P | M2GNN | .561 | .638 | .578 | .521 | .540 | .651 | .571 | .481 |
| | P | GIE | .573 | .665 | .600 | .523 | .552 | .660 | .580 | .498 |

Table 5: Statistics of five USTP datasets in NYC and CHI.

| Dataset | NYC | | | | | CHI | | | | |
|---|---|---|---|---|---|---|---|---|---|---|
| | Taxi | Bike | Mobility | Crime | 311 service | Taxi | Bike | Mobility | Crime | 311 service |
| # of records | 1,118,584 | 383,919 | 1,052,232 | 389,551 | 3,141,153 | 3,826,868 | 1,085,690 | 939,543 | 202,291 | 1,821,949 |
| # of vertices | 260 | 2,500 | 1,600 | 260 | 260 | 77 | 1,500 | 1,000 | 77 | 77 |
| Time span | 04/01/2020-06/31/2020 | | | 01/01/2021-31/12/2021 | | 04/01/2019-06/31/2019 | | | 01/01/2021-31/12/2021 | |
| Time interval | 30 minutes | | | 120 minutes | | 30 minutes | | | 120 minutes | |

## 3.3 Results

The embedding dimension is set to $d = 32$ for all methods. The result in Table 4 indicates that almost all non-Euclidean (*i.e.*, hyperbolic and spherical space) embedding methods outperform Euclidean embedding methods, aligning with their capability to explicitly represent hierarchical and cyclic structures in urban knowledge graphs. Moreover, models that derive a product space (*i.e.*, combination of hyperbolic space and spherical space) to simultaneously capture hierarchies and cycles obtain the dominant performance. This can be attributed to the fact that UrbanKGs often exhibit both high-order hierarchies and cycles, making them suitable for simultaneous modeling in hyperbolic space and spherical space. We report detailed results, analysis and implement details in Appendix A.2.

## 4 Knowledge-enhanced Urban SpatioTemporal Prediction

In this section, we first give a formal problem statement of USTP, then introduce how to incorporate UrbanKG to enhance five USTP tasks. Finally, we extensively discuss the experimental results.

### 4.1 Problem Formulation

Let $G = (V, E, A)$ denote a spatial network (*e.g.*, road network, sensor network, grid *etc.*), where $V$ and $E$ denote the set of vertices and edges. We use $A$ to denote the adjacency matrix of the spatial network $G$. We further define the graph signal matrix $X_G^{(t)} \in \mathbb{R}^{N \times C}$ for $G$, where $C$ denotes the dimension of feature, $N = |V|$ is the number of vertices, and $X_G^{(t)}$ represents the observations of spatio network $G$ at time step $t$. In general, the knowledge-enhanced urban spatiotemporal task aims to learn a multi-step prediction function $f$ based on past observations and the UrbanKG $\mathcal{G}$,

$$(X_G^{(t)}, X_G^{(t+1)}, \ldots, X_G^{t+\tau}) = f((X_G^{(t-T)}, X_G^{(t-T+1)}, \ldots, X_G^{t-1})), \mathcal{G}) \tag{1}$$

where $\mathcal{G}$ is the UrbanKG we constructed, $T$ is the input length of past observations, $\tau$ is the future steps we aim to predict.

### 4.2 Incorporating UrbanKG Embedding

We demonstrate the effectiveness of UrbanKG by directly concatenating the learned UrbanKG embeddings with the graph signal matrix. In this work, we consider the following five USTP tasks.

**Taxi Service Prediction.** Taxi service prediction aims to predict future taxi inflow and outflow based on the historical taxi service. Following existing studies [38], we split the city into $N$ disjoint areas, and connect the spatial network $G$ based on area adjacency. Let $C_{taxi}$ denote the two-dimensional feature vector (inflow and outflow). We incorporate the UrbanKG by concatenating the area embeddings with the taxi flow features, $X_G^{(t)} \in \mathbb{R}^{N \times C_{taxi'}}$, where $C_{taxi'} = e_{Area} \,||\, C_{taxi}$ and $||$ denotes the concatenation operation.

**Bike Trip Prediction.** Bike trip prediction aims to predict future inflow and outflow based on the historical bike trip. We sample $N$ roads and connect the spatial network $G$ based on road adjacency [13]. Let $C_{bike}$ represent the two-dimensional feature vector (inflow and outflow). The road embedding $e_{Road}$ is concatenated with it for UrbanKG fusion, $X_G^{(t)} \in \mathbb{R}^{N \times C_{bike'}}$, where $C_{bike'} = e_{Road} \,||\, C_{bike}$.

**Human Mobility Prediction.** Human mobility prediction aims to predict future human inflow and outflow based on historical human mobility. Following existing studies [39], We choose $N$ POIs and obtain the spatial network $G$ based on their adjacency. Let $C_{human}$ denote the two-dimensional feature vector (inflow and outflow). We integrate the UrbanKG by concatenating human mobility flow features with POI embedding, $X_G^{(t)} \in \mathbb{R}^{N \times C_{human'}}$, where $C_{human'} = e_{POI} \,||\, C_{human}$.

**Crime Prediction.** Crime prediction is a binary prediction task based on the observation of historical crime events. Following existing studies [40], we split the city into $N$ disjoint areas and connect the spatial network $G$ according to area adjacency. Let $C_{crime}$ denote the label of whether the crime occurs. We concatenate the Area embedding $e_{Area}$ with crime feature, $X_G^{(t)} \in \mathbb{R}^{N \times C_{crime'}}$, where $C_{crime'} = e_{Area} \,||\, C_{crime}$.

**311 Service Prediction.** 311 service prediction [41] is also a binary prediction task based on the observation of historical 311 complaint events. We obtain the spatial network $G$ in the same way as crime prediction. Let $C_{service}$ denote the label of the 311 complaint occurrence. We concatenate the Area embedding $e_{Area}$ with its feature, $X_G^{(t)} \in \mathbb{R}^{N \times C_{service'}}$, where $C_{service'} = e_{Area} \,||\, C_{service}$.

### 4.3 Experimental Setup

The result in Table 4 indicates that structure-aware non-Euclidean space embedding models can generate a better urban knowledge representation compared with Euclidean methods. To further verify modeling those high-order structures in Section 2.3 could enhance USTP tasks, we compare performance gaps for USTP tasks across different prediction time steps (3, 6, and 9) when using embeddings derived from different spaces. Next, we inject the embeddings from the space that yielded the greatest improvement into all downstream tasks and USTP models to confirm their benefits.

**Data Description.** The data statistics of five datasets are summarized in Table 5, and we provide the details of each dataset in Appendix A.3.

**Evaluation Metrics.** We utilize MAE and RMSE for regression tasks for evaluation [42] and use Micro-F1 and Macro-F1 [43] for classification tasks for evaluation.

**Methods.** We compare our approach with the following 9 methods: (1) Gate Recurrent Units (GRU) [44]; (2) Auto-Encoder (AE) [45]; (3) Long Short-Term Memory (LSTM) [46]; (4) SpatioTemporal Graph Convolution Network (STGCN) [47]; (5) Attention Based Spatial-Temporal Graph Convolutional Networks (ASTGCN) [48]; (6) Temporal Graph Convolutional Network (TGCN) [49]; (7) Multivariate Time Graph Neural Networks (MTGNN) [50]; (8) Adaptive Graph Convolutional Recurrent Network (AGCRN) [51]; (9) Hierarchical Graph Convolution Networks (HGCN) [52]. We incorporate UrbanKG embeddings from different spaces with USTP models as our approach, such as ASTGCN w/P and P denotes embedding from product space.

**Implement Details.** We split data with a ratio of 7:1:2 into training sets, validation sets and test sets. We use historical 12 time steps to predict the future 1 to 12 time steps. We provide implement details and more detailed results in Appendix A.4.

### 4.4 Results

**Comparison of UrbanKG Embedding Space Variants.** As depicted in Figure 4, we observe that the UrbanKG embeddings could enhance USTP tasks regardless of which spaces they come from.

Table 6: Performance gain of product space UrbanKG embedding on five spatiotemporal prediction tasks. All experimental results are the mean of 5 independent experiments repeated and the standard deviation (Std) are reported.

| Model | NYC | | | | | CHI | | | | |
|---|---|---|---|---|---|---|---|---|---|---|
| | Taxi MAE/RMSE | Bike MAE/RMSE | Mobility MAE/RMSE | Crime Micro/Macro-F1 | 311 service Micro/Macro-F1 | Taxi MAE/RMSE | Bike MAE/RMSE | Mobility MAE/RMSE | Crime Micro/Macro-F1 | 311 service Micro/Macro-F1 |
| GRU | 1.802/3.587 | 0.920/1.115 | 1.003/1.318 | -/- | -/- | 3.372/8.895 | 1.431/2.619 | 1.157/1.800 | -/- | -/- |
| LSTM | 1.425/2.769 | 0.832/0.994 | 0.984/1.227 | -/- | -/- | 3.075/9.076 | 1.359/2.259 | 1.107/1.682 | -/- | -/- |
| AE | 1.399/2.484 | 0.701/0.898 | 0.956/1.233 | 56.42/52.12 | 65.68/59.64 | 3.313/11.64 | 1.416/2.706 | 1.148/1.721 | 57.49/57.27 | 54.67/55.34 |
| STGCN | 0.835/2.069 | 0.221/0.558 | 0.553/0.976 | 76.80/63.10 | 81.15/78.50 | 1.967/6.014 | 0.632/1.537 | 0.458/1.138 | 71.86/68.71 | 79.67/79.47 |
| STGCN w/P | 0.781/1.942 | 0.201/0.543 | 0.542/0.947 | 77.56/64.26 | 81.98/79.15 | 1.844/5.826 | 0.625/1.501 | 0.414/1.064 | 72.18/68.93 | 79.92/79.83 |
| Std | 0.001/0.000 | 0.006/0.002 | 0.001/0.000 | 0.001/0.000 | 0.001/0.001 | 0.003/0.002 | 0.006/0.012 | 0.004/0.001 | 0.008/0.003 | 0.002/0.003 |
| Improv % | 6.47%/6.14% | 9.05%/2.69% | 1.99%/2.97% | 0.99%/1.84% | 1.02%/0.83% | 6.25%/3.13% | 1.11%/2.34% | 9.61%/6.50% | 0.45%/0.32% | 0.31%/0.45% |
| MTGNN | 1.289/2.275 | 0.967/1.041 | 0.963/1.163 | 72.88/58.61 | 79.32/75.75 | 2.167/5.965 | 1.153/1.723 | 1.081/1.504 | 68.48/66.46 | 76.72/76.32 |
| MTGNN w/P | 1.183/2.118 | 0.920/1.021 | 0.891/1.083 | 73.47/60.02 | 80.28/77.85 | 1.975/5.882 | 1.083/1.641 | 0.983/1.399 | 71.73/68.65 | 78.29/77.99 |
| Std | 0.007/0.005 | 0.016/0.009 | 0.021/0.011 | 0.008/0.004 | 0.011/0.009 | 0.023/0.017 | 0.005/0.009 | 0.016/0.007 | | 0.000/0.001 |
| Improv % | 8.22%/6.90% | 4.86%/1.92% | 7.48%/6.8% | 0.81%/2.41% | 1.21%/2.77% | 8.86%/1.39% | 6.07%/4.76% | 9.07%/6.98% | 4.75%/3.30% | 2.05%/2.19% |
| AGCRN | 1.315/2.391 | 0.958/1.038 | 0.945/1.148 | 65.98/60.62 | 75.68/69.64 | 2.403/7.277 | 1.187/1.768 | 0.213/1.281 | 64.14/63.35 | 71.18/67.07 |
| AGCRN w/P | 1.208/2.221 | 0.875/0.983 | 0.901/1.067 | 67.75/61.88 | 76.35/72.57 | 2.255/6.945 | 1.093/1.621 | 0.205/1.258 | 66.51/65.27 | 72.87/70.54 |
| Std | 0.016/0.021 | 0.035/0.011 | 0.006/0.005 | 0.000/0.000 | 0.001/0.003 | 0.019/0.026 | 0.007/0.011 | 0.004/0.015 | 0.001/0.000 | 0.000/0.000 |
| Improv % | 8.14%/7.11% | 8.66%/5.30% | 4.66%/7.06% | 2.68%/2.08% | 0.89%/4.21% | 6.16%/4.56% | 7.92%/8.31% | 3.76%/1.80% | 3.71%/3.03% | 2.37%/5.17% |
| ASTGCN | 0.832/2.141 | 0.231/0.579 | 0.566/0.973 | 76.69/61.62 | 80.82/77.45 | 1.849/5.323 | 0.745/1.822 | 0.505/1.259 | 71.77/68.45 | 79.18/78.93 |
| ASTGCN w/P | 0.805/2.012 | 0.220/0.564 | 0.558/0.944 | 76.94/62.34 | 81.28/78.85 | 1.822/5.198 | 0.690/1.754 | 0.463/1.191 | 72.22/68.90 | 80.09/79.98 |
| Std | 0.004/0.005 | 0.003/0.003 | 0.000/0.006 | 0.009/0.004 | 0.007/0.009 | 0.001/0.006 | 0.017/0.003 | 0.015/0.009 | 0.012/0.009 | 0.008/0.007 |
| Improv % | 3.25%/6.03% | 4.76%/2.59% | 1.41%/2.98% | 0.33%/1.17% | 0.57%/1.81% | 1.46%/2.35% | 7.38%/3.73% | 8.32%/5.40% | 0.63%/0.66% | 1.15%/1.33% |
| TGCN | 1.701/3.198 | 0.337/0.685 | 0.654/1.118 | 75.22/63.19 | 77.56/72.96 | 2.112/6.645 | 1.127/2.583 | 0.702/1.665 | 70.62/66.41 | 77.89/77.57 |
| TGCN w/P | 1.578/2.887 | 0.319/0.664 | 0.635/1.054 | 76.01/63.65 | 79.29/73.47 | 1.997/6.502 | 1.052/2.341 | 0.677/1.586 | 71.26/68.38 | 78.49/78.25 |
| Std | 0.012/0.024 | 0.009/0.007 | 0.007/0.014 | 0.000/0.000 | 0.001/0.000 | 0.013/0.035 | 0.015/0.013 | 0.009/0.011 | 0.000/0.000 | 0.000/0.000 |
| Improv % | 7.23%/9.72% | 5.34%/3.07% | 2.91%/5.72% | 1.05%/0.73% | 2.23%/0.70% | 5.45%/2.15% | 6.65%/9.37% | 3.56%/4.74% | 0.91%/2.97% | 0.77%/0.88% |
| HGCN | 1.337/2.285 | 0.951/1.138 | 0.971/1.200 | 76.04/62.26 | 75.68/69.64 | 2.765/7.849 | 1.159/1.794 | 0.661/1.277 | 67.57/62.31 | 74.67/73.53 |
| HGCN w/P | 1.282/2.124 | 0.879/1.036 | 0.921/1.104 | 76.70/63.25 | 77.41/72.76 | 2.609/7.781 | 1.092/1.682 | 0.637/1.146 | 69.17/65.7 | 75.87/75.32 |
| Std | 0.027/0.019 | 0.016/0.009 | 0.017/0.028 | 0.003/0.002 | 0.009/0.005 | 0.013/0.035 | 0.028/0.031 | 0.006/0.024 | 0.007/0.005 | 0.003/0.004 |
| Improv % | 4.11%/7.05% | 7.57%/8.96% | 5.15%/8.00% | 0.87%/1.59% | 2.29%/4.48% | 5.64%/0.87% | 5.78%/6.24% | 3.63%/10.2% | 2.37%/5.44% | 1.61%/2.43% |

Additionally, hyperbolic and spherical space embeddings demonstrate greater performance improvements compared to Euclidean methods (*i.e.*, Euclidean and complex space). Notably, the embedding derived from the product space yields the most substantial benefit for both urban flow forecasting and urban event prediction. Such result is consistent with their state-of-art performance in link prediction task. It indicates that explicitly representing urban structures is helpful for urban knowledge extraction and can be further explored to develop downstream tasks performances.

**Comparison of USTP Task Variants.** Table 6 shows the results of forecasting urban flow (*i.e.*, taxi service, bike trip and mobility prediction) for future period of 30 mins and predicting urban events (*i.e.*, urban crime and 311 service prediction) for future period of 120 mins. As can be seen, integrating product space UrbanKG embeddings consistently improves the accuracy

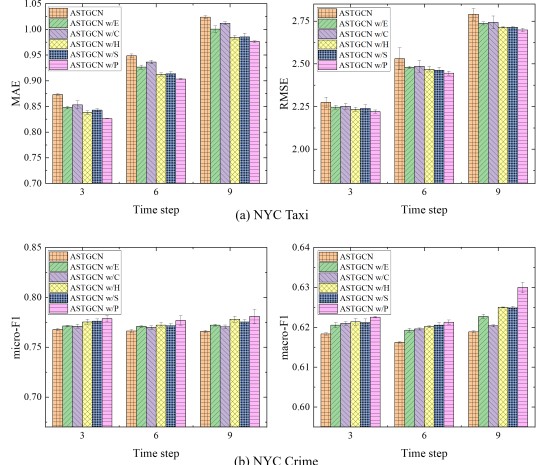

Figure 4: USTP performance comparison when incorporating embeddings derived by the best method of each space. (a) Performance on NYC Taxi. (b) Performance on NYC Crime.

of five tasks. The urban knowledge graph shows great potential (2%-10% improvement) in spatiotemporal flow prediction tasks, and it can also enhance (1%-5% improvement) ) urban event prediction. It is worth mentioning that the embeddings derived from UrbanKG are task-agnostic, where the downstream task is completely unknown. Such results indicate the UrbanKG embedding successfully captures general urban knowledge which is transferable among different USTP tasks.

**Comparison of USTP Model Variants.** It can be observed that integrating UrbanKG embeddings consistently improves the accuracy of six existing USTP models. The above results demonstrate well-extracted urban knowledge can still enhance existing USTP models (*e.g.*, ASTGCN and HGCN) from the feature representation scenario although they may already have sufficiently complex modules.

We provide implement details in Appendix A.4 and report more detailed results in Appendix A.5.

## 5    Related Work

**Domain Knowledge Graph Construction**. Knowledge graph has been widely studied in both academia and industry. In recent years, many open-source knowledge graphs have been released to

improve applications in various domains. For example, CKGG [53], OpenBG [54], BioKG [55] and recent proposed ProteinKG25 [56] are used for education, business, biomedical and protein research field respectively. However, there is still no open-source knowledge graph for urban spatiotemporal prediction tasks in the urban computing field.

**Knowledge Graph Embedding**. Knowledge graph embedding aims to project entities and relations into low-dimensional vectors, which has been widely used for various KG-based applications. Overall, the knowledge graph embedding method can be categorized into two classes, the Euclidean embedding method and the non-Euclidean embedding method. The Euclidean embedding method derives KG embeddings in the Euclidean space. For example, TransE [30], TransH [57], TransR [58], RESCAL [59], DistMult [32], TuckER [35] and ConvE [60] utilize translation, matrix decomposition, or neural network to model entities and relations and derive embeddings. The non-Euclidean embedding method aims to capture more complicated structures in the latent space. For example, MuRP [16], AttH [17], ConE [18] and UltraE [61] model hierarchies in the hyperbolic or ultrahyperbolic space. Recently, several mixed curvature embedding methods [26, 37, 62] have been proposed to simultaneously encode diverse structures (*e.g.*, hierarchies and cycles) in a product space (*i.e.*, products of constant-curvature spaces). The above structure-aware non-Euclidean methods show great potential for modeling various high-order structures in UrbanKG.

**Knowledge-enhanced Urban Spatiotemporal Prediction**. Knowledge graph has been proven useful in various urban tasks, such as traffic flow prediction [63, 64, 65, 66], mobility prediction [7, 67, 68], site selection [69], city profiling [70, 71, 72, 73] and so on [74]. Their common approach involves manually constructing a UrbanKG, and then obtaining embeddings using classical methods like TransE [30] or TuckER [35]. Although some very recent methods [70, 71, 72] design task-relevant module to capture more granular urban knowledge, their UrbanKGs are still designed for specific tasks and are publicly not available, which discourages researchers from adopting it for their own work as building an UrbanKG from scratch is time-consuming and labor-intensive. In fact, several recent works [75, 8] are striving towards this problem. However, they only describe a UrbanKG construction scheme or system but do not offer an open-source UrbanKG.

## 6 Conclusion and Limitation

In this work, we proposed UUKG, a unified dataset for knowledge-enhanced urban spatiotemporal prediction. To the best of our knowledge, UUKG is the first open-source urban knowledge graph dataset compatible with various aligned USTP tasks in the urban computing field. Extensive experimental results demonstrate the universal advancement of UUKG in improving diverse USTP tasks. Additionally, we also prove the benefits of adopting state-of-art structure-aware UrbanKG embeddding methods to further leverage rich high-order structures in UUKG.

Our work provides a unified UrbanKG construction framework but it has limitations in terms of the dataset generalizability, as all experiments were conducted in two US metropolises. In addition, we only consider concatenation operation for embedding fusion. It will be an interesting topic that how to suitably incorporate the learned embeddings to further enhance the performance of the urban spatiotemporal prediction. Despite the above limitations, we hope the constructed dataset can foster more extensive urban knowledge graph research and broad smart city application.

## 7 Maintenance Plan and Future Work

To ensure ease of access and future maintenance, the dataset and models evaluated in our paper are hosted on websites and cloud-based storage platforms. Specifically, we have created the UUKG homepage including detailed dataset descriptions and the evaluation code. The full dataset is available at Google Drive.

As an emerging building block, urban knowledge graph provides critical knowledge for urban spatiotemporal prediction models. Inevitably, more dataset, benchmarks and tools will be needed to facilitate its potential achievement. Therefore, we plan to continuously update UUKG by adding new urban entities derived from different data sources, new downstream tasks, and more friendly data usage toolkits and model interfaces. UUKG currently only contains the urban structural dataset for five types of USTP tasks. In the future, we will derive extra multi-modal data (*e.g.*, images, reviews) to enrich the UrbanKG and introduce more USTP tasks (*e.g.*, trajectory prediction and site selection).

## Acknowledgments and Disclosure of Funding

This research was supported in part by the National Natural Science Foundation of China under Grant No.62102110, the Alibaba Innovative Research Program (AIR). We appreciate the guidance provided by the team of Mr. Wang Qing, who is the Regional General Manager of Alibaba Cloud, in the Chongqing City Brain project.

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
