# A Appendix

## A.1 Datasheet for UrbanKG Dataset

We have presented the UrbanKG dataset construction progress in Section 2. The detailed statistics of entities of UrbanKG are shown in Table 7. As can be seen, we define 15 categories of POI including finance, parking area, shopping, catering, and so on. In addition, we preserve six-types of most frequent road segments including motorway (expressway or river crossing tunnels), primary traffic road, secondary traffic road, tertiary traffic road, residential traffic road, and trunk (branch roads such as expressway outbound bypass roads). We also keep five-types of frequent road junction including motorway junction (road junction in expressway or river-crossing tunnel), traffic signal (road junction having traffic light), turning circle (road junction which is a roundabout), stop (road junction having stop signal) and crossing (road junction with no special type).

## A.2 UrbanKG Embedding Implement Details and Detailed Results

We implement all models by using PyTorch. All experiments are conducted on eight NVIDIA RTX 3090 GPUs. We use Adam [76] as the optimizer and the negative sampling size is fixed to 50. We conduct a grid search to select learning rate in $\{0.001, 0.005, 0.01, 0.05, 0.1\}$ and batch size in $\{512, 1024, 2048, 4096, 5120\}$. The best hyper-parameters are selected by early stop on the validation sets and we report them in Table 8. The performance of different models in high dimensional setting ($d = 150$) is similar with our analysis in Section 3.3 and we report the detailed reuslts in Tabel 9.

## A.3 Datasheet for USTP Dataset

### A.3.1 Dataset Construction

In this section, we detail the datasets construction process of five USTP tasks step by step.

**Taxi Service**: (1) Data collection: NYC TCL[6] provides yellow taxi trip records which could cover 260 areas in NYC, thus we choose it to construct our dataset. A taxi trip record contains pick-up time, pick-up location, drop-off time and drop-off locations. For example, *[2020/4/1/00:15, (40.72, -73.98), 2020/4/1/09:10, (40.73, -73.97)]* is a taxi trip record. (2) Dataset construction: the inflow and outflow of an area can be calculated by counting the amount of taxis entering and leaving within a period of time. For NYC taxi service prediction, we count the inflow and outflow in each area every 30 minutes from 1st April to 31st June, 2020. For CHI taxi service prediction, we count the inflow and outflow in each area every 30 minutes from 1st April to 31st June, 2019. By the above steps, we obtain the spatio-temporal dataset of taxi service.

**Bike Trip**: (1) Data collection: Amazon S3[7] provides NYC bike trip data, which records the start and end points of the bike trip and has precise latitude and longitude. Every record contains the start time and end time. For example, *[2020/4/1/4:20, (40.68, -74.01), 2020/4/1/4:26, (40.68, -73.99)]* is a bike trip record. (2) Dataset construction: the inflow and outflow of a road can be calculated by counting the amount of bikes entering and leaving within a period of time. Due to the lack of bike flow data for most roads at each time step, we filter and sample roads to prevent excessive long tail issues. Specifically, for NYC bike trip prediction, we calculate the inflow and outflow on each road at 30-minute intervals from April 1st to June 31st, 2020. Subsequently, we select 2,500 roads from those with the highest bike flow values. For CHI bike trip prediction, we count the inflow and outflow in each road every 30 minutes from 1st April to 31st June, 2019. Next, we select 1,500 roads from those with the highest bike flow values. Through these steps, we obtain the spatio-temporal datasets for bike trip prediction.

**Human Mobility**: We construct human mobility dataset based on taxi service and bike trip data. The inflow and outflow of a POI can be calculated by counting the number of taxi passengers, taxi drivers and bikers, who enter and leave within a period of time.

---

[6]https://www.nyc.gov/site/tlc/about/tlc-trip-record-data.page
[7]https://s3.amazonaws.com/tripdata/index.html

Table 7: Information for category of POI, Road and Junction in UrbanKG.

| Entity | Description | Quantity NYC | Quantity CHI |
|---|---|---|---|
| PC | catering | 501 | 280 |
| | corporation | 821 | 1,042 |
| | culture and education | 2,296 | 2,352 |
| | domestic service | 155 | 15 |
| | finance | 112 | 64 |
| | governments and organization | 538 | 143 |
| | medical and health | 337 | 237 |
| | parking area | 31,484 | 10,545 |
| | place of workshop | 1,592 | 1,057 |
| | public service | 1,609 | 1,141 |
| | residential area | 21,166 | 9,885 |
| | scenic spot | 164 | 802 |
| | shopping | 1,362 | 2,690 |
| | sports and leisure | 143 | 885 |
| | transportation | 185 | 450 |
| RC | motorway | 2,523 | 1,011 |
| | primary | 7,173 | 4,289 |
| | secondary | 12,717 | 13,024 |
| | tertiary | 10,964 | 7,385 |
| | residential | 77,024 | 45,775 |
| | trunk | 524 | 100 |
| JC | crossing | 45,182 | 32,158 |
| | motorway junction | 704 | 250 |
| | stop | 1,437 | 155 |
| | traffic signal | 14,677 | 3,902 |
| | turning circle | 441 | 626 |

Due to the absence of mobility flow data for most POIs at each time step, we filter and sample POIs to avoid excessive long tail issues. For NYC mobility prediction, we calculate the inflow and outflow at each POI at 30-minute intervals from April 1st to June 31st, 2020. Then, we choose 1,600 roads from those with the highest mobility flow values. For CHI mobility prediction, we measure the inflow and outflow at each POI at 30-minute intervals from April 1st to June 31st, 2019. Subsequently, we select 1,000 POIs from those with the highest mobility flow values. By the above steps, we obtain the spatio-temporal dataset of POI-level mobility prediction.

**Crime**: (1) Data collection: NYC OpenData[8] provides crime occurrence data, which includes all valid felony, misdemeanor and violation crimes reported to the New York City Police Department (NYPD). Every crime record contains crime type, time, latitude and longitude. For example, *[FELONY, 2021/12/17/22:13, (40.64, -73.90)]* is a crime record. (2) Dataset construction: the crime label for an area can be established by examining if any criminal activities occur within a designated time frame. Due to the sparsity of crime occurrence, we create the crime label in each area every 120 minutes from 1st Jan to 31st Dec, 2021. By the above steps,

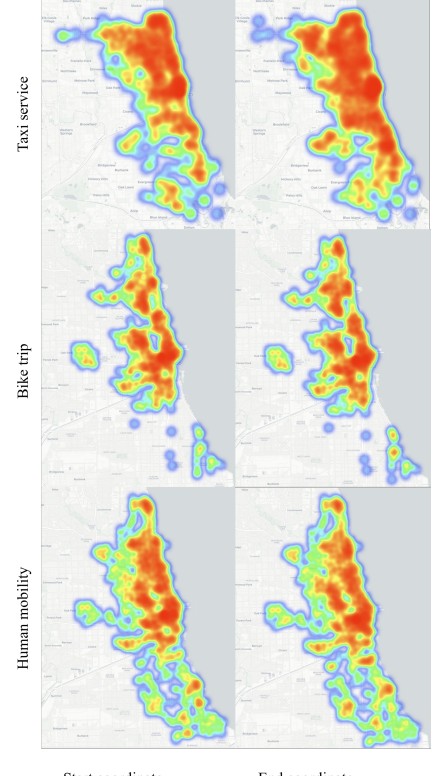

Figure 5: Heat map of the start and end coordinate of taxi, bike, and human trip in CHI USTP dataset.

[8]https://data.cityofnewyork.us/Public-Safety/NYPD-Complaint-Data-Historic/qgea-i56i

Table 8: Best hyperparameters of different UrbanKG embedding methods.

| Space | Model | Learning rate | Optimizer | Batch size | Negative samples |
|---|---|---|---|---|---|
| Euclidean models | TransE | 0.001 | Adam | 2048 | 50 |
| | DisMult | 0.0005 | Adam | 2048 | 50 |
| | MuRE | 0.001 | Adam | 4096 | 50 |
| | TuckER | 0.0005 | Adam | 1024 | 50 |
| | RotatE | 0.05 | Adagrad | 2048 | 50 |
| | CompIEx | 0.05 | Adagrad | 2048 | 50 |
| | QuatE | 0.001 | Adam | 2048 | 50 |
| Non-Euclidean models | MuRS | 0.001 | Adam | 4096 | 50 |
| | MuRP | 0.001 | Adam | 4096 | 50 |
| | RotH | 0.001 | Adam | 4096 | 50 |
| | RefH | 0.05 | Adagrad | 4096 | 50 |
| | AttH | 0.001 | Adam | 4096 | 50 |
| | ConE | 0.001 | Adam | 4096 | 50 |
| | M2GNN | 0.001 | Adam | 4096 | 50 |
| | GIE | 0.001 | Adam | 4096 | 50 |

Table 9: Overall link prediction results for high-dimensional embeddings $d = 150$.

| Type | Space | Model | NYC | | | | CHI | | | |
|---|---|---|---|---|---|---|---|---|---|---|
| | | | MRR | Hits@10 | Hits@3 | Hits@1 | MRR | Hits@10 | Hits@3 | Hits@1 |
| Euclidean models | E | TransE | .547 | .592 | .553 | .501 | .517 | .582 | .544 | .475 |
| | E | DisMult | .462 | .514 | .477 | .402 | .432 | .499 | .485 | .426 |
| | E | MuRE | .556 | .632 | .581 | .513 | .524 | .642 | .565 | .462 |
| | E | TuckER | .532 | .627 | .584 | .492 | .513 | .602 | .547 | .466 |
| | C | RotatE | .302 | .395 | .342 | .265 | .344 | .412 | .367 | .291 |
| | C | ComplEx | .289 | .388 | .337 | .253 | .352 | .417 | .365 | .297 |
| | C | QuatE | .355 | .421 | .386 | .322 | .431 | .514 | .458 | .396 |
| Non-Euclidean models | S | MuRS | .566 | .647 | .585 | .506 | .538 | .649 | .570 | .471 |
| | H | MuRP | .559 | .639 | .591 | .512 | .533 | .643 | .569 | .477 |
| | H | RotH | .571 | .654 | .594 | .511 | .546 | .648 | .573 | .481 |
| | H | RefH | .562 | .636 | .581 | .509 | .539 | .642 | .572 | .477 |
| | H | ATTH | .573 | .656 | .592 | .513 | .545 | .647 | .571 | .489 |
| | H | ConE | .565 | .637 | .588 | .511 | .538 | .639 | .566 | .485 |
| | P | M2GNN | .578 | .642 | .601 | .523 | .552 | .647 | .582 | .503 |
| | P | GIE | .587 | .651 | .607 | .532 | .561 | .656 | .596 | .513 |

we obtain the spatio-temporal dataset of crime prediction.

**311 Service**: (1) Data collection: NYC 311 [9] provides and updates daily 311 service request automatically, which includes air quality issue, illegal parking and so on. Every 311 service record contains complaint type, created date, latitude and longitude. For example, *[Illegal parking, 2021/06/27/12:42, (40.88, -73.89)]* is a 311 service record. (2) Dataset construction: the 311 service label for an area can be obtained by examining if any 311 service activities occur within a designated time frame. We create the 311 service label in each area every 120 minutes from 1st Jan to 31st Dec, 2021. By the above steps, we obtain the spatio-temporal dataset of 311 service prediction.

### A.3.2 Dataset Visualization

In addition, we take Chicago as an example to visualize the spatial and temporal distribution of various USTP dataset and the pattern in New York is similar. As shown in Figure 5, we can see the start points and end points of the three urban trip (*i.e.*, taxi service, bike trip, human mobility, crime and 311 service). Figure 6 also shows the spatial distribution of crime events and 311 service complaints. We illustrates the temporal distribution of the these five tasks in Figure 7.

---

[9]https://portal.311.nyc.gov/

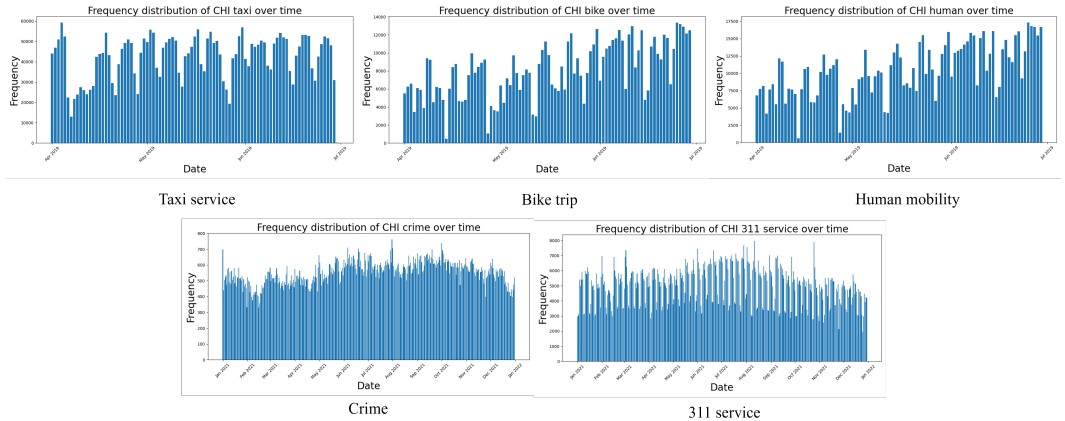

| Taxi service | Bike trip | Human mobility |
| --- | --- | --- |

| Crime | 311 service |
| --- | --- |

Figure 7: Frequency histogram of taxi, bike, human trip, crime and 311 service event in CHI USTP dataset.

## A.4 USTP Models Implement Details

We split data with a ratio of 7:1:2 into training sets, validation sets and test sets. We use historical 12 time steps to predict the future 1 to 12 time steps. We implement all the models and methods in pytorch with eight NVIDIA RTX 3090 GPUs. We use Adam [76] as the optimizer and the batch size is fixed to 64. The best model hyperparameters are determined using early stopping on the validation sets. The final configurations for each model are saved as JSON files in our open-source code https://github.com/usail-hkust/UUKG/.

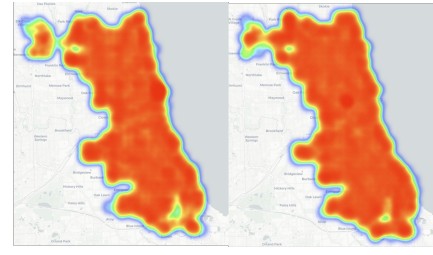

| Crime | 311 service |
| --- | --- |

Figure 6: Heat map of the event coordinate of crime and 311 service.

## A.5 USTP Detailed Results

Section 4 has demonstrates effectiveness of product space UrbanKG embedding in improving diverse USTP tasks, this section provides more detailed results. Specifically, we compare the impact on downstream USTP task performance when product space embeddings of different dimensions (*i.e.*, 8, 16, 32, 64 and 128 dimensions) are used for downstream tasks.

**Comparison of UrbanKG Embedding dimensions.** We report the results of forecasting NYC taxi flow and NYC crime events for furture 3, 6 and 9 time step. As shown in Figure 8, we present a comparison of performance gaps for USTP tasks using product embeddings of different dimensions. It is evident that the UrbanKG embedding improves USTP performance across a range of embedding dimensions from 8 to 128. Moreover, the performance improvement becomes less significant when the embedding dimension exceeds 32.

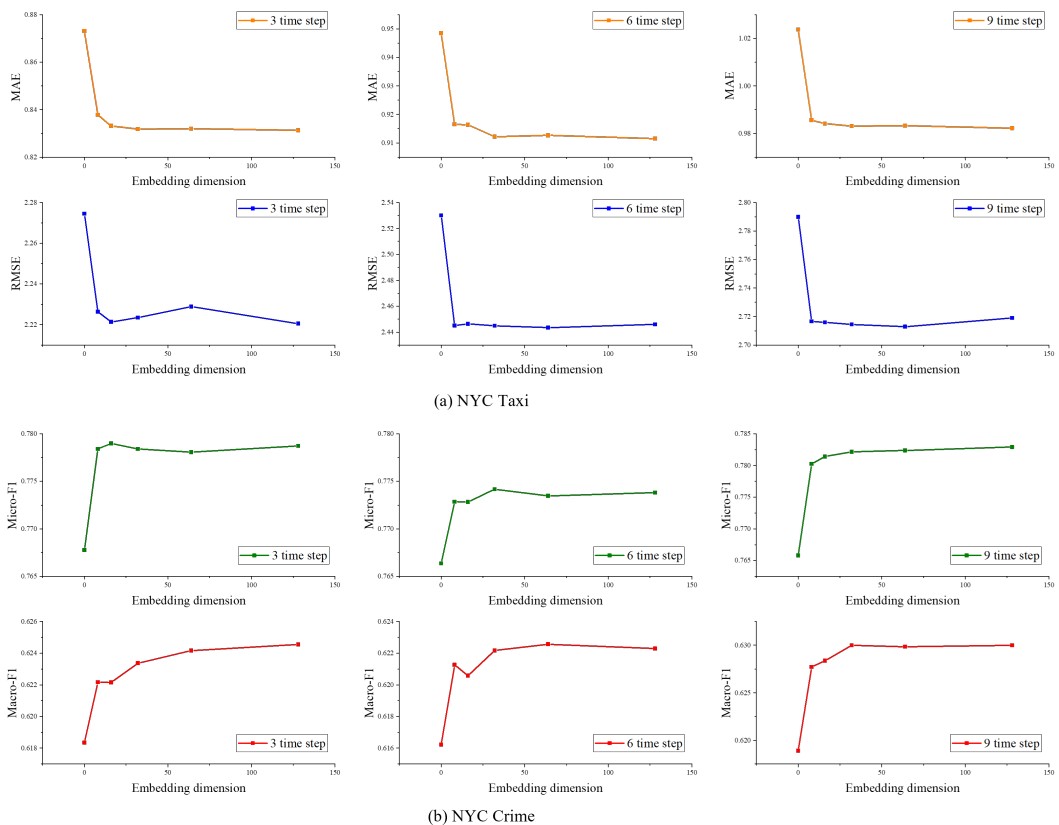

Figure 8: USTP performance comparison when incorporating product space embeddings with different dimension into ASTGCN model. The performance of embedding dimension '0' is the result is the ASTGCN model without UrbanKG embedding. (a) USTP Performance on NYC Taxi. (b) USTP Performance on NYC Crime.