# OpenReview forum: "UUKG: Unified Urban Knowledge Graph Dataset for Urban Spatiotemporal Prediction"
_NeurIPS.cc/2023/Track/Datasets_and_Benchmarks — NeurIPS 2023 Datasets and Benchmarks Poster_

### Official Review · Reviewer_6kYe · 2023-07-19

**Rating:** 7
**Confidence:** 4
**Correctness:** Yes.
**Clarity:** Yes.

**Strengths:**

1. The paper is easy to follow and well-written. The idea of using urban knowledge graphs to improve downstream tasks is interesting.
2. Extensive knowledge-graph embedding methods are implemented. The empirical results show that product space-based methods obtain dominant performance.
3. The improvements of incorporating learned KG embeddings are significant on five downstream USTP tasks, which validates the effectiveness of the embeddings and opens new research opportunities.


**Additional Feedback:**

No.

**Documentation:**

No, the data documentation can be enhanced.

**Ethics:**

No.

**Limitations:**

Yes.

**Opportunities For Improvement:**

My major concerns are the dataset accessibility and documentation:
1. It seems the authors only release part of the dataset on GitHub, and the complete data is actually not available in google drive.
2. The licenses of code and dataset are not mentioned in the paper.
3. There is no mention of a dataset maintenance plan in the paper.
4. In the GitHub repository, the instructions to preprocess the data, and to reproduce the experimental results are not clear, and very short. This is not user-friendly.

Some minor comments:
1. Please unify the expressions of "spatiotemporal" and "spatio-temporal".
2. In line 99, what is the platform to query the corresponding road network data?
3. In line 109, the explanation of coordinate may be placed at the first occurrence of the word, i.e., line 102.
4. The caption of Table 3 says the hyperbolicity values should be always greater then zero, but the values are actually zeros in the table.
5. In line 233, how to build the adjacency matrix?

**Relation To Prior Work:**

Yes.

**Summary And Contributions:**

This paper proposes UrbanKGs, a unified urban knowledge graph dataset for knowledge-enhanced urban spatiotemporal predictions. The dataset consists of millions of triplets from two metropolises in US, namely New York and Chicago. To validate the usage of UrbanKGs, the authors implement 15 KG embedding methods, evaluate them on the KG completion task, and integrate the learned embeddings into spatio-temporal models for five downstream forecasting tasks. The extensive results demonstrate the effectiveness of KG embeddings.

---

> ### Author Response · Authors · 2023-08-18
> **Response to Reviewer 6kYe**
>
> **Comment:** We highly appreciate the reviewer recognized that our work is interesting and could open new research opportunities for the community. We also thanks for the reviewer's very high-quality and constructive suggestions which will assist us in further revising our paper. Please find the point-to-point response to the reviewer's comments below.
>
> > [OFI-1, OFI-2 & OFI-4] It seems the authors only release part of the dataset on GitHub, and the complete data is actually not available in google drive. The licenses of code and dataset are not mentioned in the paper. In the GitHub repository, the instructions to preprocess the data, and to reproduce the experimental results are not clear, and very short. This is not user-friendly.
>
> **[Response]** The complete dataset (**13.71GB** in total) is now available on [Google Drive](https://drive.google.com/drive/folders/1egTmnKRzTQuyW_hsbFURUonGC-bJmBHW?usp=sharing). The dataset includes all meta data, preprocessed data, and final datasets used for UrbanKG embedding training and USTP tasks training. Following the [MIT license](https://github.com/usail-hkust/UUKG/blob/main/LICENSE.txt), one can conduct re-development based on their needs. In our updated [GitHub repository](https://github.com/usail-hkust/UUKG), the detailed dataset explanation, guidance of data usage and preprocessing are provided. As for the reproduction, researchers can utilize our dataset and reproduce the experimental results following the implement details provided in both supplementary material (section A.2, section A.4 and Table 8) and [config JSON files](https://github.com/usail-hkust/UUKG/tree/main/USTP_Model/libcity/config/model/traffic_state_pred).
>
> > [OFI-3] There is no mention of a dataset maintenance plan in the paper.
>
> **[Response]** Thanks to the reviewer's kind remind. We have updated the GitHub repository with detailed description of project maintenance plan. For the long-term maintenance of UUKG, we will keep integrating more urban entities, new downstream tasks and more toolkits to the framework. We have added a new section in our revision paper to discuss the maintenance issue and please feel free to check the updated version.
>
> > [Some minor comments] 1.Please unify the expressions of "spatiotemporal" and "spatio-temporal". 2. In line 99, what is the platform to query the corresponding road network data? 3. In line 109, the explanation of coordinate may be placed at the first occurrence of the word, i.e., line 102. 4. The caption of Table 3 says the hyperbolicity values should be always greater then zero, but the values are actually zeros in the table. 5. In line 233, how to build the adjacency matrix?
>
> **[Response]**
> 1. Thanks for valuable suggestion and we have unified this term as "spatiotemporal" in our revision paper.
> 2. Sorry for confusion. Same with query step with POI data, we collect road network data from Open Street Map (OSM). We have provided further explanation (Line 100 in our revision paper) of the data acquisition process in our paper.
> 3. Thanks for your valuable suggestion. We have moved the explanation of 'coordinate' into its first occurrence (Line 103 in our revision paper).
> 4. Sorry for this minor mistake. The hyperbolicity values should be always greater or equal to zero. We have fixed this error.
> 5. Sorry for confusion. We utilize the spatial distance between two vertices in the spatial network to construct a weighted adjacency matrix. The adjacency matrix example can be found in our [GitHub](https://github.com/usail-hkust/UUKG/tree/main/USTP_Model/raw_data). We have provided more details on the construction of adjacency matrix in our revisied supplementary material (Line 651, 665, 678, 697, 708).
>
> We highly appreciate the reviewer's detailed suggestions, which helps us improve the quality our paper. And will further polish the paper before camera ready if paper accepted.

---

> > ### Comment · Reviewer_6kYe · 2023-08-18
> >
> > Thank you very much for solving my concerns and for your efforts on preparing the detailed response. I updated my score to 7.

---

### Official Review · Reviewer_E5fz · 2023-07-19
**Review of Submission 807**

**Rating:** 6
**Confidence:** 4
**Clarity:** Yes. The flow of the paper is good.

**Strengths:**

1. This paper qualitatively and quantitatively analyzes the diverse high-order structures ((i.e., hierarchies and cycles) in UrbanKG dataset.
2. The evaluation experiments that include implementation of KG embedding methods and integrating KG embeddings into spatiotemporal models, which shows the importance of urban high-order structure modeling.


**Additional Feedback:**

None

**Correctness:**

The claims make in the paper are correct and the dataset is constructed in a sound way. The benchmark evaluation methods and experiment design are appropriate and performed correctly.

**Documentation:**

The documentation is sufficient.

**Ethics:**

I don’t see any ethical concerns.

**Limitations:**

The authors adequately don’t address the limitations. This paper should consider more operations such as addition, multiplication, etc. to integrate UrbanKG embedding into the Urban spatiotemporal prediction.

**Opportunities For Improvement:**

1. Only some of the examples in UrbanKG dataset are found at https://github.com/usail-hkust/UUKG/. The complete dataset is not available.
2. As I know, there are two ways to integrate KG embeddings. One is to pre-train KG embeddings and then integrate them into downstream tasks, and the other is to train directly with downstream tasks. But this paper is not clear about how to train and integrate KG Embedding.
3. This paper has incorrect sentence description. For example, “The results are reported in Table 4” in 3.3 section and “The result in Table 4 indicates that…” in 4.3 section are inconsistent.


**Relation To Prior Work:**

The coverage of prior work is sufficient.

**Summary And Contributions:**

This paper proposes an open-source urban knowledge graph (UrbanKG) for knowledge-enhanced urban spatiotemporal prediction. This urban knowledge graph dataset uncovers diverse high-order structural patterns, such as hierarchies and cycles. The authors evaluate the proposed UrbanKG dataset using 15 KG embedding methods and integrate the learned KG embeddings into 9 spatiotemporal models. Based on evaluation, the authors highlight the potential of state-of-the-art high-order structure-aware UrbanKG embedding methods.

---

> ### Author Response · Authors · 2023-08-18
> **Response to Reviewer E5fz**
>
> **Comment:** We highly appreciate your high-quality review and valuable suggestions. Please find a point-to-point response to the reviewer's comments below.
>
> > [OFI-1] Only some of the examples in UrbanKG dataset are found. The complete dataset is not available.
>
> **[Response]** The complete dataset (Total **13.71 GB**) now is available on [Google Drive](https://drive.google.com/drive/folders/1egTmnKRzTQuyW_hsbFURUonGC-bJmBHW?usp=sharing). Following the MIT license, one can conduct their research on our dataset.
>
> > [OFI-2] As I know, there are two ways to integrate KG embeddings. One is to pre-train KG embeddings and then integrate them into downstream tasks, and the other is to train directly with downstream tasks. But this paper is not clear about how to train and integrate KG Embedding.
>
> **[Response]** Sorry for the confusion. Our motivation is to verify all types of entity embeddings can enhance downstream tasks. Following a simple yet effective routine, we directly concatenate learned KG embedding in link prediction task with spatiotemporal features (e.e., taxi flow and mobility flow) as the input of downstream tasks (Line 246, Line 252 and Line 258). We agree with the reviewer that pretraining or joint learning may further unleash the predictive power of our KG Embeddings and worth to be investigated in the future. We will further highlight the design choice in the final version to avoid confusion.
>
> > [OFI-3] This paper has incorrect sentence description. For example, “The results are reported in Table 4” in 3.3 section and“The result in Table 4 indicates that…” in 4.3 section are inconsistent.
>
> **[Response]** We appreciate the reviewer for pointing out the inconsistent expression in our work, and we have made the corrections (Line 221 and 260 in our revision paper) and carefully check the paper for other potential typos and mistakes.
>
> > [L-1] The authors adequately don’t address the limitations. This paper should consider more operations such as addition,multiplication, etc. to integrate UrbanKG embedding into the Urban spatiotemporal prediction.
>
> **[Response]** We sincerely appreciate the potential limitation provided by the reviewer. It will be an interesting topic that how to suitably incorporate the learned embeddings to further enhance the performance of the urban spatiotemporal prediction. We have included the limitation into our revised paper (Line 348-350) and please feel free to check the update version.

---

> ### Author Response · Authors · 2023-08-25
> **Response to Reviewer E5fz**
>
> Dear reviewer E5fz:
>
> We sincerely appreciate precious review time and valuable comments. Following your suggestions, we have revised our paper and provided corresponding responses. Please let us know if you still have any unclear parts of our work.
>
> Best,
>
> NeurIPS 2023 Track Datasets and Benchmarks Submission 807 Authors

---

> > ### Comment · Reviewer_E5fz · 2023-08-26
> >
> > Thanks for your careful responses to my concerns. I will keep my score as 6 and tend to accept this paper.

---

### Official Review · Reviewer_fgmx · 2023-07-20
**Some parts still need to be improved and clarified**

**Rating:** 6
**Confidence:** 4
**Correctness:** Correct
**Clarity:** Great

**Strengths:**

1. Clear paper structure, and the paper is easy to follow.

2. Open source datasets.

3. The dataset is about a novel topic that may contribute to the practical scenarios of KGs.

**Additional Feedback:**

See Strengths and Limitations

**Documentation:**

There is sufficient detail to support reproducibility.

**Limitations:**

The paper presents great efforts in constructing the Urban KGs. However, the good quality of the proposed dataset is essential for researchers to present a fair and valuable evaluation of their methods. In other words, the novel benchmark dataset is even more important than a novel method. It will guide the direction and quality of new models in this area since the researchers will regard it as the benchmark and use it for evaluation. From this point of view, the following 4 points can be improved:

1. Lack of comparison between the proposed Urban KGs with existing Urban KGs, such as [1].

     [1] Liu, Yu, et al. "UrbanKG: An Urban Knowledge Graph System." ACM Transactions on Intelligent Systems and Technology 14.4 (2023): 1-25.

2. More baseline models should be compared, especially for existing hyper-graph models for urban computing, such as [2][3][4][5].

     [2] Li, Zhonghang, et al. "Spatial-temporal hypergraph self-supervised learning for crime prediction." 2022 IEEE 38th International Conference on Data Engineering (ICDE). IEEE, 2022.

     [3] Zhang, Qianru, et al. "Automated Spatio-Temporal Graph Contrastive Learning." Proceedings of the ACM Web Conference 2023.

     [4] Sun, Yanfeng, et al. "Dual dynamic spatial-temporal graph convolution network for traffic prediction." IEEE Transactions on Intelligent Transportation Systems 23.12 (2022): 23680-23693.

     [5] Wang, Chenyu, et al. "Hagen: Homophily-aware graph convolutional recurrent network for crime forecasting." Proceedings of the AAAI Conference on Artificial Intelligence. Vol. 36. No. 4. 2022.

3. The datasets split the train : valid: test as 18:1:1 for both NYC and CHI. I am wondering why the authors select this ratio. Meanwhile, the link prediction results are still not that good with such a split ratio. Does it indicate the dataset is not of good quality? Besides, more current KGE models should be evaluated, such as CoMPGCN [2]. More compared models can be found in related surveys [6][7][8].

     [6] Vashishth, Shikhar, et al. "Composition-based multi-relational graph convolutional networks." arXiv preprint arXiv:1911.03082 (2019).

     [7] Ji, Shaoxiong, et al. "A survey on knowledge graphs: Representation, acquisition, and applications." IEEE transactions on neural networks and learning systems 33.2 (2021): 494-514.

     [8] Liang, Ke, et al. "Reasoning over different types of knowledge graphs: Static, temporal and multi-modal." arXiv preprint arXiv:2212.05767 (2022).

4. Missing some related works. As for the KG construction, more information can be referred to in [9]. Some GNN based KGE models are not mentioned (refering to [7][8])

     [9] Zhong, Lingfeng, et al. "A comprehensive survey on automatic knowledge graph construction." arXiv preprint arXiv:2302.05019 (2023).

**Opportunities For Improvement:**

See Limitations

**Relation To Prior Work:**

Ok, but still can be improved.

**Summary And Contributions:**

The paper provides a new Urban KG for CHI and NYC, an open-source KG for smart city and urban computing, which contribute to the community.

---

> ### Author Response · Authors · 2023-08-18
> **Response to Reviewer fgmx [1/2]**
>
> **Comment:**
>
> We appreciate the reviewer recognized our paper is well-written, the open-sourced datasets is valuable for the community. We also thank the reviewer's detailed suggestions and questions. Please find the point-to-point response to the reviewer's comments below.
>
> > [L1] Lack of comparison between the proposed Urban KGs with existing Urban KGs.
>
> **[Response]**  Sorry for misunderstanding. In fact, we have discussed the work [1] mentioned by the reviewer (<u>The 71th reference in our paper</u>) in the related work section (<u>Line 336 - Line 338</u>). As mentioned in the paper, the pioneer work [1] only describes a UrbanKG construction system or scheme but didn't not offer an open-source UrbanKG. In contrast, we open source two large-scale UrbanKG datasets as well as the UrbanKG construction framework, which is one of the major contributions of this work.
>
> Furthermore, another technical distinction of our work lies in the innovative exploration of state-of-the-art non-Euclidean space embedding models to derive structure-aware UrbanKG embeddings. These embeddings have been demonstrated to yield further improvements in a range of downstream urban spatiotemporal forecasting tasks.
>
> [1] Liu, Yu, et al. "UrbanKG: An Urban Knowledge Graph System." ACM Transactions on Intelligent Systems and Technology 14.4 (2023): 1-25.
>
>
> > [L2] More baseline models should be compared, especially for existing hyper-graph models for urban computing, such as [2-5].
>
> **[Response]**  We appreciate the reviewer introduced more knowledge-enhanced spatiotemporal prediction works [2-5], and we will be further discussed them the related work. Nevertheless, the authors wish to clarify that the above works are all designed to solve a specific predictive task and can not be easily generalized to other spatiotemporal prediction problems. In fact, this is one major concern that we choose ST-GCN or ASTGCN as the base model.
>
> Specifically, [2] designs dedicated crime embedding layer, type-aware spatial pattern encoding, and specific temporal crime dependency module to better solve the crime prediction task and [5] explicitly use POI statistical features and homophily ratio for crime prediction. Such frameworks are tailored to address crime prediction and could not be extended to other spatiotemporal prediction tasks such as the taxi, bike and mobility prediction. Therefore, it is unapplicable to adopt them as baselines. It is in a same situation for [3] and [4]. particularly, [3] constructs multiple view-graph (e.g., POI-view, trajectory-view and distance-view) to help crime, taxi and bike trip prediction. The external data used (such as trajectory) and problem definition in this paper are quite different from ours, thus it is also hard to directly adopt it as a baseline. [4] is tailored to solve road-level traffic flow prediction, thus it could not be extended to crime prediction or region-level taxi flow prediction in our paper.
>
> Although the above methods [2-5] are not suitable to be used as baselines, we acknowledge they all belong to the knowledge-enhance urban spatial-temporal prediction. We will add a further discussion about them in the related work section.
>
> [2] Li, Zhonghang, et al. "Spatial-temporal hypergraph self-supervised learning for crime prediction." 2022 IEEE38th International Conference on Data Engineering (ICDE). IEEE, 2022.
> [3] Zhang, Qianru, et al. "Automated Spatio-Temporal Graph Contrastive Learning." Proceedings of the ACM Web Conference 2023.
> [4] Sun, Yanfeng, et al. "Dual dynamic spatial-temporal graph convolution network for traffi c prediction." IEEE Transactions on Intelligent Transportation Systems 23.12 (2022): 23680-23693.
> [5] Wang, Chenyu, et al. "Hagen: Homophily-aware graph convolutional recurrent network for crime forecasting."Proceedings of the AAAI Conference on Artificial Intelligence. Vol. 36. No. 4. 2022.

---

> ### Author Response · Authors · 2023-08-18
> **Response to Reviewer fgmx [2/2]**
>
> > [L3-1] The datasets split the train : valid: test as 18:1:1 for both NYC and CHI. I am wondering why the authors select this ratio. Meanwhile, the link prediction results are still not that good with such a split ratio. Does it indicate the dataset is not of good quality?
>
> **[Response]** Thanks to the reviewer's question to help us clarify poential confusion. In this work, we split 10% of our dataset into equal validation and test sets (i.e., train:valid:test as 18:1:1). Similar split ratio is commonly used by the knowledge graph research community on various popular KG datasets [6-8] (e.g., 7.11% in WN18RR and 13.96% in FB15K-237).
>
> Moreover, we apologize for the confusion on dataset quality. In fact, link prediction is a commonly used task to evaluate the learned KG embedding quality (<u>Line 200 -  Line 202</u>) rather than dataset quality. Take other widely used high-quality KG datasets (e.g., WN18RR and FB15K-237) for example, the state-of-the-art link prediction results [9-10]  on WN18RR (with an MRR indicator of around 0.47) and on FB15K-237 (with an MRR indicator of around 0.334) are also with large room to be improved. Following their task setting, we also use link prediction to evaluate the learned UrbanKG embedding on NYC and CHI, and to help identify suitable models for UrbanKG representations.
>
> [6] Sun, Zhiqing, et al. "Rotate: Knowledge graph embedding by relational rotation in complex space."  *Proceedings of International Conference on Learning Representations 2019*. ICLR 2019.
> [7] Dettmers, Tim, et al. "Convolutional 2d knowledge graph embeddings." *Proceedings of the AAAI conference on artificial intelligence*. AAAI 2018.
> [8] Toutanova, Kristina, et al. "Representing text for joint embedding of text and knowledge bases." *Proceedings of the 2015 conference on empirical methods in natural language processing*. EMNLP 2015.
> [9] Nayyeri, Mojtaba, et al. "Knowledge Graph Embeddings using Neural Ito Process: From Multiple Walks to Stochastic Trajectories." *Findings of the Association for Computational Linguistics: ACL 2023*. 2023.
> [10] Gregucci, Cosimo, et al. "Link prediction with attention applied on multiple knowledge graph embedding models." *Proceedings of the ACM Web Conference 2023*. 2023.
>
> > [L3-2 & L4] Besides, more current KGE models should be evaluated, such as CoMPGCN [11].More compared models can be found in related surveys.
> > Missing some related works. As for the KG construction, more information can be referred to in [14]. Some GNN based KGE models are not mentioned (refering to [12-13])
>
> **[Response]** We appreciate the reviewer introduced more works [11-14] related to knowledge graph. The mentioned surveys [12-14] primarily related to automatic KG construction, temporal KG, multi-model KG and KG representation. Their research focuses are different with ours, i.e., construct and release UrbanKG datasets in a reproducible routine and access how to obtain UrbanKG embedding by leveraging state-of-art non-Euclidean space embedding models. Actually, we have compared various methods (e.g., TransE, TuckER, RotatE, CompIEx, QuatE, MuRP and AttH) mentioned in the above surveys [12-14]. By considering the extensive workload, we plan to compare more latest methods in the next stage.
>
> Moreover, GNN-based methods such as CompGCN [11] and R-GCN [15] usually deal with small and moderate sized KG with no more than 120 thousand entities like YAGO3-10 [16], which is computationally uncontractable on our large-scale constructed UrbanKGs. Thus we removed GNN aggregation layer when implementing M2GNN [17] to reduce the computational overhead (Line 238 in our revision paper). We left how to apply GNN-based methods on large-scale UrbanKGs a future research direction.
>
> We will further clarify each baseline and explain the rational of baseline selection in our paper.
>
> [11] Vashishth, Shikhar, et al. "Composition-based multi-relational graph convolutional networks." arXiv preprintarXiv:1911.03082 (2019).
> [12] Ji, Shaoxiong, et al. "A survey on knowledge graphs: Representation, acquisition, and applications." IEEE transactions on neural networks and learning systems 33.2 (2021): 494-514.
> [13] Liang, Ke, et al. "Reasoning over diff erent types of knowledge graphs: Static, temporal and multi-modal." arXivpreprint arXiv:2212.05767 (2022).
> [14] Zhong, Lingfeng, et al. "A comprehensive survey on automatic knowledge graph construction." arXiv preprintarXiv:2302.05019 (2023).
> [15] Schlichtkrull, Michael, et al. "Modeling relational data with graph convolutional networks." ESWC. 2018.
> [16] Mahdisoltani, F., et al. Yago3: A knowledge base from multilingual wikipedias. CIDR. 2015.
> [17] Wang, Shen, et al. "Mixed-curvature multi-relational graph neural network for knowledge graph completion." *Proceedings of the Web Conference 2021*. 2021.

---

> ### Author Response · Authors · 2023-08-25
> **Response to Reviewer fgmx**
>
> Dear reviewer fgmx:
>
> We thank you for the precious review time and valuable comments. We have carefully considered your comments and have provided corresponding responses, which we believe have covered your concerns. We hope to further discuss with you whether or not your concerns have been addressed. Please let us know if you still have any unclear parts of our work.
>
> Best,
>
> NeurIPS 2023 Track Datasets and Benchmarks Submission 807 Authors

---

> > ### Comment · Reviewer_fgmx · 2023-08-28
> >
> > Thank you for your response to my concerns. Most of the concerns are addressed.

---

### Official Review · Reviewer_SDLE · 2023-07-21

**Rating:** 7
**Confidence:** 4
**Correctness:** Yes
**Clarity:** Yes

**Strengths:**

1. This paper presents the first open-sourced Urban Knowledge Graph for spatial-temporal prediction.
2. The experiments show that adding the learned knowledge graph embeddings for downstream tasks (e.g., bike flow prediction, crime prediction) could improve model's performance.
3. The writing is clear in general.

**Additional Feedback:**

N/A

**Documentation:**

Yes.

**Limitations:**

1. The setting in Sec. 4.2 is a little bit confusing. Do you concatenate the learned KG embeddings for inputs only?
2. It is suggested that the authors add std for Table 6 and Figure 4 to better understand the variance.

**Opportunities For Improvement:**

Please see the limitations.

**Relation To Prior Work:**

Yes

**Summary And Contributions:**

This paper introduces a unified urban knowledge graph dataset (UUKG) for knowledge enhanced urban spatiotemporal predictions. It first constructs a large-scale urban knowledge graph (UrbanKGs), and then conduct qualitative and quantitative analysis on UrbanKGs for many downstream Urban SpatioTemporal Prediction (USTP) tasks.

---

> ### Author Response · Authors · 2023-08-18
> **Response to Reviewer SDLE**
>
> **Comment:**
>
> We highly appreciate your high-quality review and valuable suggestions. Please find a point-to-point response to the reviewer's comments below.
>
> > [L1] The setting in Sec. 4.2 is a little bit confusing. Do you concatenate the learned KG embeddings for inputs only?
>
> **[Response]**  Sorry for the confusion. Our motivation is to verify all types of entity embeddings can enhance downstream tasks. Following a straightforward yet effective routine, we directly concatenate learned KG embedding with the input features. Note another advanced approaches to integrate the KG embeddings are also applicable. We will further highlight the design choice in the final version to avoid confusion.
>
> > [L2] It is suggested that the authors add std for Table 6 and Figure 4 to better understand the variance.
>
> **[Response]**  Thank you for your invaluable suggestion. We concur that the standard deviation (std) serves as a crucial indicator for evaluating the stability of a model's performance. Therefore, the std for each experiment was recorded when our experiments were conducted independently with random seeds, repeated 5 times. In our revision paper, we have added the std to the Table 6, as well as updated the std bars in Figure 4 accordingly. Most of our experimental standard deviations are in the negative cubic level, indicating a relative stability. Please feel free to check the revised version in our paper.

---

### Official Review · Reviewer_QJBt · 2023-07-22
**A good dataset paper with certain limitations**

**Rating:** 7
**Confidence:** 3
**Correctness:** Yes.
**Clarity:** Yes.

**Strengths:**

The authors have effectively elucidated the significance of such a dataset in advancing research in this domain. The technical construction is robust.

**Additional Feedback:**

It would be better if the authors can have more discussions about the limitations.

**Documentation:**

Yes.

**Ethics:**

No.

**Limitations:**

It is worth noting a potential area for improvement regarding the authors' claim of the dataset being "unified." As the dataset includes only two US cities, it would be better to acknowledge the limitation that the findings may not generalize to cities in other countries or of different scales.

**Opportunities For Improvement:**

It is worth noting a potential area for improvement regarding the authors' claim of the dataset being "unified." As the dataset includes only two US cities, it would be better to acknowledge the limitation that the findings may not generalize to cities in other countries or of different scales.

**Relation To Prior Work:**

Yes.

**Summary And Contributions:**

This paper introduces UUKG, an urban knowledge graph dataset designed to enhance knowledge in urban spatiotemporal predictions. The authors have effectively elucidated the significance of such a dataset in advancing research in this domain. They have constructed urban KGs, comprising millions of triplets, for two major metropolises by connecting 10 diverse urban entities. Their experimental results demonstrate that exploiting high-order structural patterns can yield valuable insights for downstream urban spatial temporal prediction tasks. The paper includes benchmarks and detailed implementation discussions provided in the Appendix.

Overall, the motivation behind the research is well-founded, and the technical construction is robust.

However, it is worth noting a potential area for improvement regarding the authors' claim of the dataset being "unified." As the dataset includes only two US cities, it would be better to acknowledge the limitation that the findings may not generalize to cities in other countries or of different scales.

---

> ### Author Response · Authors · 2023-08-18
> **Response to Reviewer QJBt**
>
> **Comment:**
>
> We appreciate the reviewer recognized that our problem is well-motivated and technical construction is  comprehensive. We also thank the reviewer's valuable suggestions.
>
> > [L1] It is worth noting a potential area for improvement regarding the authors' claim of the dataset being "unified." As the dataset includes only two US cities, it would be better to acknowledge the limitation that the findings may not generalize to cities in other countries or of different scales.
>
> **[Response]**  We sincerely appreciate the reviewer's valuable advice to help us improve the quality of the paper. As suggested, we have included the limitation (Line 346-347) in the revised paper. Due to page limit, we also introduce how to create a customizing UrbanKG dataset in our [GitHub repositories](https://github.com/usail-hkust/UUKG). Researchers can build UrbanKGs for other cities based on our framework.

---

### Decision · Program_Chairs · 2023-09-22

**Decision:**

Accept (Poster)

**Comment:**

All the reviewers have appreciated the work and given good scores. I too concur with them and recommend acceptance.